# Neural Unsigned Distance Fields for Implicit Function Learning

**Julian Chibane**      **Aymen Mir**      **Gerard Pons-Moll**

Max Planck Institute for Informatics, Saarland Informatics Campus, Germany
`{jchibane,amir,gpons}@mpi-inf.mpg.de`

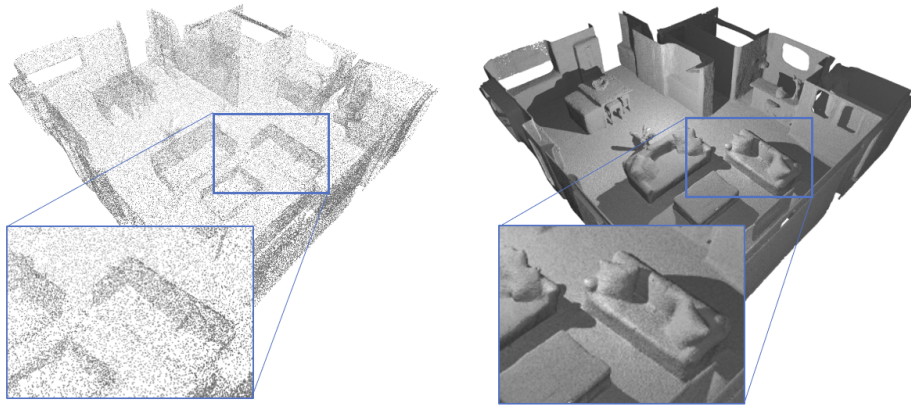

Figure 1: Our method can represent and reconstruct complex open surfaces. Given a sparse test point cloud of a captured room (left) it generates a detailed, completed scene (right).

## Abstract

In this work we target a learnable output representation that allows continuous, high resolution outputs of arbitrary shape. Recent works represent 3D surfaces implicitly with a Neural Network, thereby breaking previous barriers in resolution, and ability to represent diverse topologies. However, neural implicit representations are *limited to closed* surfaces, which divide the space into inside and outside. Many real world objects such as walls of a scene scanned by a sensor, clothing, or a car with inner structures are not closed. This constitutes a significant barrier, in terms of data pre-processing (objects need to be artificially closed creating artifacts), and the ability to output open surfaces. In this work, we propose *Neural Distance Fields* (NDF), a neural network based model which *predicts the unsigned distance* field for arbitrary 3D shapes given sparse point clouds. NDF represent surfaces at high resolutions as prior implicit models, but do not require closed surface data, and significantly broaden the class of representable shapes in the output. NDF allow to extract the surface as very dense point clouds and as meshes. We also show that NDF allow for surface normal calculation and can be rendered using a slight modification of sphere tracing. We find NDF can be used for multi-target regression (multiple outputs for one input) with techniques that have been exclusively used for rendering in graphics. Experiments on ShapeNet [13] show that NDF, while simple, is the state-of-the art, and allows to reconstruct shapes with inner structures, such as the chairs inside a bus. Notably, we show that NDF are not restricted to 3D shapes, and can approximate more general *open* surfaces such as curves, manifolds, and functions. Code is available for research at [1].

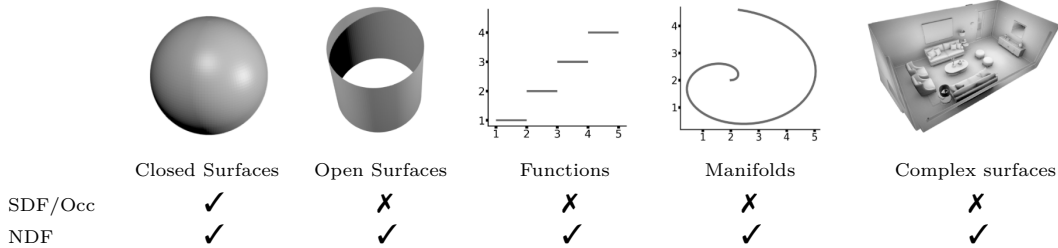

| | Closed Surfaces | Open Surfaces | Functions | Manifolds | Complex surfaces |
|---|---|---|---|---|---|
| SDF/Occ | ✓ | ✗ | ✗ | ✗ | ✗ |
| NDF | ✓ | ✓ | ✓ | ✓ | ✓ |

Figure 2: Recent works rely on occupancies or signed distances to represent surfaces, which limits shapes to be closed. In NDF, we learn with an un-signed distance field representation, allowing us to reconstruct a broader class of shapes.

# 1 Introduction

Reconstructing continuous and renderable surfaces from unstructured and incomplete 3D point-clouds is a fundamental problem in robotics, vision and graphics. The choice of shape representation is central for effective learning. There have been a growing number of papers on implicit function learning (IFL) for shape representation and reconstruction tasks [53, 14, 49, 68, 15, 27, 11, 10]. The idea is to train a neural network which classifies continuous points in 3D as *inside* or *outside* the surface via occupancies or signed distance fields (SDF). Compared to point, mesh or voxels-based methods, IFL can output continuous surfaces of arbitrary resolution and can handle more topologies.

A major limitation of existing IFL approaches is that they can only output *closed* surfaces – that is, surfaces which divide the 3D space into inside and outside the surface. Many real world objects such as cars, cylinders, or a wall of a scanned 3D scene can not be represented. This is a barrier, both in terms of tedious data pre-processing — surfaces need to be closed which often leads to artifacts and loss of detail — and more importantly the ability to output open surfaces.

In this paper, we introduce *Neural Distance Fields* (NDF), which do not suffer from the above limitations and are a more general shape representation for learning. NDF directly predict the *unsigned* distance field (UDF) to the surface – we regress, for a point $\mathbf{p} \in \mathbb{R}^d$, the distance to the surface $\mathcal{S} \subset \mathbb{R}^d$ with a learned function $f(\mathbf{p}) : \mathbb{R}^d \mapsto \mathbb{R}_0^+$ whose zero-levelset $f(\mathbf{p}) = 0$ represents the surface. In contrast to SDFs or occupancies, this allows to naturally represent surfaces which are open, or contain objects inside, like the bus with chairs inside in Figure 2. Furthermore, NDF is not limited to 3D shape representation ($d = 3$), but allows to represent functions, *open* curves and surface manifolds (we experimented with $d = 2, 3$), which is not possible when using occupancy or SDFs.

Learning with UDF poses new challenges. Several applications require extracting point clouds, meshes or directly rendering the implicit surface onto an image, which requires finding its zero-levelset. Most classical methods, such as marching cubes [48] and volume rendering [23], find the zero-levelset by detecting flips from inside to outside and vice versa, which is not possible with UDF. However, exploiting properties of UDFs and fast gradient evaluation of NDF, we introduce easy to implement algorithms to compute dense point clouds and meshes, as well as rendered images from NDF.

Experiments on ShapeNet [13] demonstrate that NDF significantly outperform the state-of-the-art (SOTA) and, unlike all IFL competitors except [5], do not require pre-computing closed meshes for training. More importantly, in comparison to *all* IFL methods (including [5]), we can represent and reconstruct shapes with inner structures and layering. To demonstrate the wide applicability of NDF beyond 3D shape reconstruction, we use them for classical regression tasks – we interpolate linear, quadratic and sinusoidal functions, as well as manifold data, such as spirals, which is not possible with occupancies or SDFs. In contrast to standard regression based on $L_2$ or $L_1$ losses which tends to average multiple modes, NDF can naturaly produce multiple outputs for a single input. Interestingly, we show that function regression $y = f(\mathbf{x})$ can be cast as sphere tracing the NDF, effectively leveraging a classical graphics technique for a core machine learning task.

In summary, we make the following contributions:

- We introduce NDF as a new representation for 3D shape learning, which in contrast to occupancies or SDFs, do not require to artificially close shapes for training, and can represent open surfaces, shapes with inner structures, and open manifolds.

- We obtain SOTA results on reconstruction from point clouds on ShapeNet using NDF.

- We contribute simple yet effective algorithms to generate dense point-clouds, normals, meshes and render images from NDF.

- To encourage further research in this new direction, we make code and model publicly available at *https://virtualhumans.mpi-inf.mpg.de/ndf/*.

## 2    Related Work

Distance fields can be found in computer vision, graphics, robotics and physics [37]. They are used for shape registration [26], model fitting [69, 3], to speed up inference in part based models [25], and for extracting skeletons and medial axis [22, 8]. However, to our knowledge, unsigned distance fields have not been used for learning 3D shapes. Here, we limit our discussion to *learning* based methods for 3D shape representation and reconstruction.

### 2.1    Learning with Voxels, Meshes and Points-Clouds

Since convolutions are natural on voxels, they have been the most popular representation for learning [39, 35, 64], but the memory footprint scales cubically with resolution, which has limited grids to small sizes of $32^3$ [44, 82, 16, 74]. Higher resolutions ($256^3$) [81, 80, 86] can be achieved at the cost of slow training, or difficult multi-resolution implementations [31, 71, 76]. Replacing occupancy with Truncated Signed Distance functions [17] for learning [19, 42, 65, 70] can reduce quantization artifacts, nonetheless TSDF values need to be stored in a grid of fixed limited resolution.

Mesh based methods deform a template [77, 63, 58] but are limited to a single topology or predict vertices and faces directly [29, 18], but do not guarantee surface continuity. Alternatively, a shape can be approximated predicting by convexes directly [20] or combining voxels and meshes [28], but results are still coarse. Meshes are common to represent humans and have been used for estimating pose, shape [38, 40, 41, 52, 75, 85] and clothing [4, 3, 12] from images, but topologies and detail are restricted by an underlying model like SMPL [47] or SMPL + Garments meshes (ClothCap [57], TailorNet [54] and GarNet [30], SIZER [73]).

For point clouds (PCs) the pioneering PointNet based architectures [60, 61] spearheaded research, such as kernel point convolutions [72, 59, 33, 78], tree-based graph convolutions [67] normalizing flows [84], combinations of points and voxels for efficiency [46] or domain adaptation techniques [62]. Due to their simplicity, PCs have been used to represent shape in reconstruction and generation tasks [24, 34, 84], but the number of output points is typically fixed beforehand, limiting the effective resolution of such methods.

### 2.2    Implicit Function Leaning (IFL)

IFL methods use either binary occupancies [49, 27, 15, 66, 21] or signed distance functions [53, 14, 50, 36] as shape representation for learning. They predict occupancy or the SDF values at *continuous* point locations (*x-y-z*). Like our model (NDF), in stark contrast to PC, voxel and mesh-based methods, IFL techniques are not limited by resolution and are flexible to represent different topologies. Unlike our NDF, they require artificially closing the shapes during a pre-processing step, which often leads to loss of detail, artifacts or lost inner structures. The recent work of [5] (SAL) does not require to close training data. However, the final output is again an SDF prediction, and hence can only represent closed surfaces. In the experiments we show that this leads to missing interior structures for cars (missing chairs, steering wheel). Moreover, NDF can be trained on multiple object classes jointly, whereas SAL diverges from the signed distance solution and fails in this setting.

To our knowledge, all existing methods can only represent *closed surfaces* because they rely on classifying the 3D space into inside and outside regions. Instead in NDF, we regress the unsigned distance to the surface. This is simple to implement, but it is a powerful

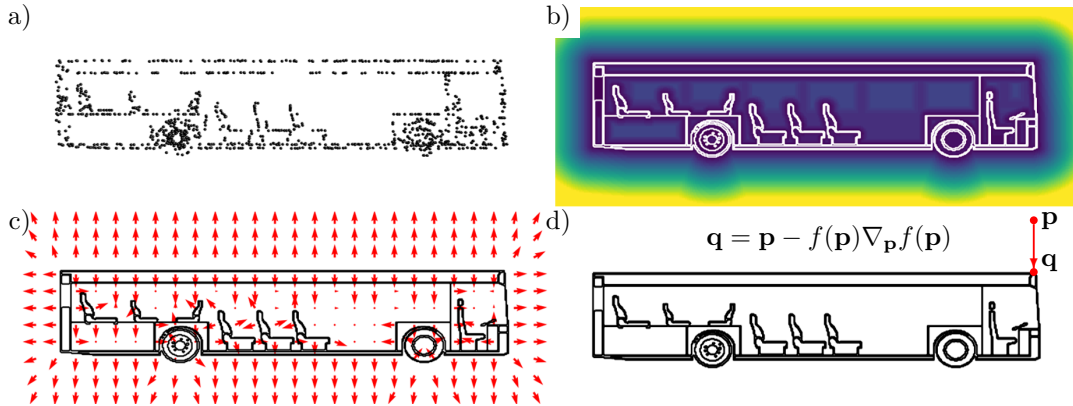

Figure 3: Overview of Point Cloud Inference, visualized on a 2D slice of a 3D bus. a) A sparse input is given. b) For each point in 3D, the unsigned distance field is predicted from the input with NDF. This yields a *continuously completed* representation of *arbitrary resolution and topology*. c) The corresponding gradient field of NDF can be elegantly computed analytically with back-propagation. Gradients pointing towards the depth direction appear as a dot. d) A point $\mathbf{p} \in \mathbb{R}^3$ in 3D space, is moved $f(\mathbf{p})$ units in the negative gradient direction $-\nabla_{\mathbf{p}} f(\mathbf{p})$ to yield its predicted closest surface point $\mathbf{q}$.

generalization which brings numerous advantages. NDF can represent any surface, closed or open, approximate data manifolds, and we show how to extract dense point-clouds, meshes and images from NDF. This enables us to complete open garment surfaces and large multi object real world 3D spaces, see experiments Sec. 4.

A second class of IFL methods focus on neural rendering, that is, on generating novel view-points given one or multiple images (SRN [68] and NeRF [51]). We do not focus on neural rendering, but on 3D shape reconstruction from deficient point clouds, and in contrast to NeRF, our model can be trained on multiple scenes or objects. Finally, it is worth mentioning that there exist connections between NDF and energy based models [43]– instead of distance a fixed energy budget is used and energy at training data points is minimized, which effectively shapes an implicit function.

## 3  Method

### 3.1  Background On Implicit Function Learning

The first IFL papers for 3D shape representation learn a function, which given a vectorized shape code $\mathbf{z} \in \mathcal{Z}$, and a point $\mathbf{p} \in \mathbb{R}^3$ predict an occupancy $f(\mathbf{p}, \mathbf{z}) \colon \mathbb{R}^3 \times \mathcal{Z} \mapsto [0, 1]$ or a signed distance function $f(\mathbf{p}, \mathbf{z}) \colon \mathbb{R}^3 \times \mathcal{Z} \mapsto \mathbb{R}$. Instead of decoding based on point locations, IF-Nets [15] achieve significantly more accurate and robust reconstructions by decoding occupancies based on features extracted at continuous locations of a multi-scale 3D grid of deep features. Hence, in NDF we use the same shape latent encoding.

### 3.2  Neural Distance Fields

Our formulation of NDF predicts the *unsigned* distance field of surfaces, instead of relying on inside and outside. This simple modification allows us to represent a much wider class of surfaces and manifolds, not necessarily closed. However, the lack of sign brings new challenges: 1) UDFs are not differentiable exactly at the surface, and 2) most algorithms to generate points, meshes and render images work with SDFs or occupancies exclusively. Hence, we present algorithms which exploit NDF to visualize the reconstructed implicit surfaces as 1) dense point-clouds and meshes (Subsec. 3.3), and 2) images rendered from NDF directly (Subsec. 3.4), including normals (Subsec. 3.4). For completeness, we first explain our shape encoding, which follows IF-Nets [15]. Thereafter we describe our shape decoding and visualization strategies.

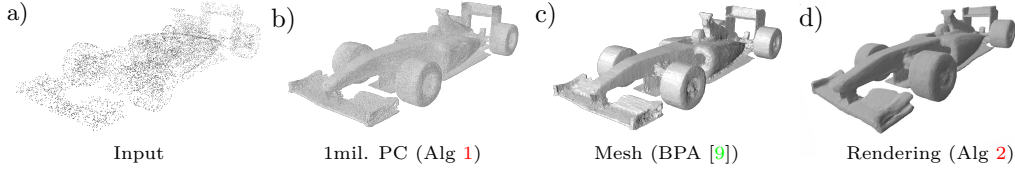

| a) | b) | c) | d) |
|---|---|---|---|
| Input | 1mil. PC (Alg 1) | Mesh (BPA [9]) | Rendering (Alg 2) |

Figure 4: Visualization techniques for NDF. Given an sparse input a), NDF produce a completed representation. We extract a dense point cloud b) using Alg. 1, meshable with off-the-self methods c) or render an image d) with an adoption of sphere tracing (Alg. 2).

**Shape Encoding:** The task is to reconstruct a continuous detailed surface $\mathcal{S}$ from a sparse point-cloud $\mathbf{X} \in \mathcal{X}$. The input is voxelized and encoded by a 3D CNN as a multi-scale grid of deep features $\mathbf{F}_1, .., \mathbf{F}_n$, $\mathbf{F}_k \in \mathcal{F}_k^{K \times K \times K}$, where the grid size $K$ varies with scale and $\mathcal{F}_k \in \mathbb{R}^C$ is a feature with multiple channels C, see IF-Nets [15].

**Shape Decoding:** The task of the decoder is to predict the unsigned distance function (UDF) for points to the ground truth surface $\mathrm{UDF}(\mathbf{p}, \mathcal{S}) = \min_{\mathbf{q} \in \mathcal{S}} \|\mathbf{p} - \mathbf{q}\|$. Let $\Psi_{\mathbf{x}}(\mathbf{p}) = (\mathbf{F}_1(\mathbf{p}), \dots, \mathbf{F}_n(\mathbf{p}))$ define a neural function $\Psi_{\mathbf{x}}(\mathbf{p}) : \mathbb{R}^3 \mapsto \mathcal{F}_1 \times \dots \times \mathcal{F}_n$ that extracts a deep feature at point $\mathbf{p}$ from the multi-scale encoding (described above) of input $\mathbf{X}$. Let $\Phi((\mathbf{F}_1(\mathbf{p}), \dots, \mathbf{F}_n(\mathbf{p}))) : \mathcal{F}_1 \times \dots \times \mathcal{F}_n \mapsto \mathbb{R}^+$ be a learned function which regresses the unsigned distance from deep features, with its layers activated by ReLU non-linearities (this ensures $\Phi \geq 0$). By composition we obtain our *Neural Distance Fields* (NDF) approximator:

$$f_{\mathbf{x}}(\mathbf{p}) = (\Phi \circ \Psi_{\mathbf{x}})(\mathbf{p}) : \mathbb{R}^3 \mapsto \mathbb{R}_0^+, \tag{1}$$

which maps points to the unsigned distance. One important aspect is that the spatial gradient $\nabla_{\mathbf{p}} f_{\mathbf{x}}(\mathbf{p})$ of the distance field can be computed analytically using standard back-propagation through NDF. When the dependency of $f_{\mathbf{x}}(\mathbf{p})$ with $\mathbf{x}$ is not needed for understanding, we will simply denote NDF with $f(\mathbf{p})$.

**Learning:** For training, pairs $\{\mathbf{X}_i, \mathcal{S}_i\}_{i=1}^T$ of inputs $\mathbf{X}_i$ with corresponding surfaces $\mathcal{S}_i$ are required. Note that surfaces can be mathematical functions, curves or 3D meshes. We create training examples by sampling points $\mathbf{p}$ in the vicinity of the surface and compute their ground truth $\mathrm{UDF}(\mathbf{p}, \mathcal{S})$. We jointly learn the encoder and decoder of the NDF $f_{\mathbf{x}}$ parameterized by neural parameters $\mathbf{w}$ (denoted $f_{\mathbf{x}}^{\mathbf{w}}$) via the mini-batch loss

$$\mathcal{L}_{\mathcal{B}}(\mathbf{w}) := \sum_{\mathbf{x} \in \mathcal{B}} \sum_{\mathbf{p} \in \mathcal{P}} |\min(f_{\mathbf{x}}^{\mathbf{w}}(\mathbf{p}), \delta) - \min(\mathrm{UDF}(\mathbf{p}, \mathcal{S}_{\mathbf{x}}), \delta)|,$$

where $\mathcal{B}$ is a input mini-batch and $\mathcal{P}$ is a sub-sample of points. Similar to learning with SDFs [53], clamping the maximal regressed distance to a value $\delta > 0$ concentrates the model capacity to represent the vicinity of the surface accurately. Larger $\delta$ values increase the convergence of our visualization Algorithms 1 and 2. We find a good trade-off with $\delta = 10$cm.

### 3.3 Dense Point Cloud Extraction from NDF

Extracting dense point clouds (PC) is necessary for applications such as point based modelling [55, 2], and is a common shape representation for learning [60]. If NDF $f(\mathbf{p})$ are a perfect approximator of the true $\mathrm{UDF}(\mathbf{p})$, we can leverage the following nice property of UDFs. A point $\mathbf{p}$ can be projected on the surface with the following equation:

$$\mathbf{q} := \mathbf{p} - f(\mathbf{p}) \cdot \nabla_{\mathbf{p}} f(\mathbf{p}), \quad \mathbf{q} \in \mathcal{S} \subset \mathbb{R}^d, \quad \forall \mathbf{p} \in \mathbb{R}^d / \mathcal{C}, \tag{2}$$

where $\mathcal{C}$ is the set of points which are equidistant to at least two surface points, referred to as the cut locus [79]. Although this projection trick did not find much use beyond early works on digital sculpting of shapes [6, 56], it is instrumental to extract the surface as points efficiently from NDF, see Fig. 4 for an example. The intuition is clear, see Fig. 3; the negative gradient points in the direction of fastest distance decrease, and consequently also points towards the closest point: $-\nabla_{\mathbf{p}} f(\mathbf{p}) = \lambda \cdot (\mathbf{q} - \mathbf{p}), \lambda \in \mathbb{R}$. Also, the norm of the gradient equals one $\|\nabla_{\mathbf{p}} f(\mathbf{p})\| = 1$. Therefore, we need to move a distance of $f(\mathbf{p})$ along the

negative gradient to reach $\mathbf{q}$. In order to transfer these useful theoretical properties to NDF, which are only an approximation to the true UDF, we address two challenges: 1) NDF have inaccuracies such that Eq. 2 does not directly produce surface points, 2) NDF are clamped beyond a distance of $\delta = 10$cm from the surface. To address 1), we found that projecting a point with Eq. 2 multiple times (we use 5 in our experiments) with unit normalized gradient yields accurate surface predictions. Each projection takes around 3.7 seconds for a 1 million points on a Tesla V100. For 2), we sample many points from a uniform distribution within the bounding box (BB) of the surface and regress their distances. We project points $\mathbf{p}$ with valid NDF distance $f(\mathbf{p}) < \delta$ to the surface. This yields only a sparse point cloud as the large majority of points will be $f(\mathbf{p}) > \delta$. To obtain a dense PC, we sample (from a $d-$Gaussian with variance $\delta/3$) around the sparse PC and project again. The procedure is detailed in (Alg. 1). Since we are able to efficiently extract millions of points, naive classical algorithms for meshing [9] (which locally connect the dots) can be used to generate high quality meshes. Because each point can be processed independently, the algorithm can be parallelized on a GPU, resulting in fast dense-point cloud generation.

---

**Algorithm 1:** NDF: Dense PCs

$\mathcal{P}_{\text{init}}$: $m$ points uniformly sampled in BB
$\mathcal{P}_{\text{init}} \leftarrow \{\mathbf{p} \in \mathcal{P}_{\text{init}} | f(\mathbf{p}) < \delta\}$
**for** $i = 1$ to num_steps **do**
$\qquad \mathbf{p} \leftarrow \mathbf{p} - f(\mathbf{p}) \cdot \frac{\nabla_{\mathbf{p}} f(\mathbf{p})}{\|\nabla_{\mathbf{p}} f(\mathbf{p})\|}, \quad \forall \mathbf{p} \in \mathcal{P}_{\text{init}}$
**end for**
$\mathcal{P}_{\text{dense}}$: $n$ points drawn with replacement from $\mathcal{P}_{\text{init}}$
$\mathcal{P}_{\text{dense}} \leftarrow \{\mathbf{p}+\mathbf{d}|\mathbf{p} \in \mathcal{P}_{\text{dense}}, \mathbf{d} \sim \mathcal{N}(0, \delta/3)\}$

**for** $i = 1$ to num_steps **do**
$\qquad \mathbf{p} \leftarrow \mathbf{p} - f(\mathbf{p}) \cdot \frac{\nabla_{\mathbf{p}} f(\mathbf{p})}{\|\nabla_{\mathbf{p}} f(\mathbf{p})\|}, \quad \forall \mathbf{p} \in \mathcal{P}_{\text{dense}}$
**end for**
**return** $\{\mathbf{p} \in \mathcal{P}_{\text{dense}} | f(\mathbf{p}) < \delta\}$

---

**Algorithm 2:** NDF: Roots along ray

$\lambda = 0; \mathbf{p} = \mathbf{p}_0;$
**while** $f(\mathbf{p}) > \epsilon_1$ **do**
$\qquad \lambda = \lambda + \alpha \cdot f(\mathbf{p});$
$\qquad \mathbf{p} = \mathbf{p}_0 + \lambda \cdot \mathbf{r};$
**end**
Refinement step using Taylor approximation;
**while** $f(\mathbf{p}) > \epsilon_2$ **do**
$\qquad \lambda = \lambda + \beta \cdot \frac{-f(\mathbf{p})}{\mathbf{r}^T \nabla_p f(\mathbf{p})};$
$\qquad \mathbf{p} = \mathbf{p}_0 + \lambda \cdot \mathbf{r};$
**end**
Approximate normal;
$\mathbf{n}(\mathbf{p}) \leftarrow \nabla_{\mathbf{p}} f(\mathbf{p} - \epsilon \cdot \mathbf{r})$

---

### 3.4 Ray Tracing to Render Surfaces and Evaluate Functions and Manifolds

**Rendering NDF:** Finding the intersection of a ray with the surface is necessary for direct image rendering. We show that a slight modification of the sphere tracing algorithm [32] is enough to render NDF images, see Figure 4. Formally, a point $\mathbf{p}$ along a ray $\mathbf{r}$ can be expressed as $\mathbf{p} = \mathbf{p}_0 + \lambda \cdot \mathbf{r}$, where $\mathbf{p}_0$ is an initial point. We want to find $\lambda$ such that $f(\mathbf{p})$ falls bellow a minimal threshold. The basic idea of sphere tracing is to march along the ray $\mathbf{r}$ in steps of size equal to the distance at the point $f(\mathbf{p})$, which is theoretically guaranteed to converge to the correct solution for exact UDFs.

Since NDF distances are not exact, we introduce two changes in sphere tracing to avoid missing the intersections (over-shooting). First, we march with damped sphere tracing $\alpha \cdot f(\mathbf{p})$, $\alpha \in (0, 1]$ until we get closer than $\epsilon_1$ from the surface $0 < \epsilon_1 < f(\mathbf{p})$. This reduces the probability of over-shooting when distance predictions are too high. However, our steps are always in ray direction. In case we still over-shoot we need to take steps backwards. This is possible near the surface $(f(\mathbf{p}) < \epsilon_1)$ by computing the zero intersection $\gamma$ along the ray solving the Taylor approximation $f(\mathbf{p}) + \gamma \cdot \mathbf{r}^T \nabla_{\mathbf{p}} f(\mathbf{p}) \approx f(\mathbf{p} + \gamma \cdot \mathbf{r}) = 0$ for $\gamma$. We then move along the ray with $\beta \cdot \gamma$, where $\beta \in (0, 1]$.

We iterate until we are sufficiently close $f(\mathbf{p}) < \epsilon_2 < \epsilon_1$, see Algorithm 2. See the supplementary for parameter values used in the experiments. *Analysis:* in absence of noise, $f(\mathbf{p})$ tends to 0 as we approach the surface: $\lim_{\mathbf{p} \to \mathbf{q}} f(\mathbf{p}) = 0, \mathbf{q} \in \mathcal{S}$, and consequently so does the damped sphere tracing step $\alpha \cdot f(\mathbf{p})$. Also, theoretically, note that at any given point $\mathbf{p}$ not on the surface $f(\mathbf{p}) > 0$, we can safely move $f(\mathbf{p})$ units along the ray without intersecting the surface, because $f(\mathbf{p})$ is the minimal distance to the surface.

**Multi-target Regression with NDF:** Consider the classical regression task $y = h(x_1, \ldots, x_n)$. When the data has multiple $y$ for $x$, learning $h(\cdot)$ from a training set

$\{\mathbf{p}_i\}_{i=0}^N$ of points $\mathbf{p}_i$ tends to average out the multiple outputs $y$. To address this, we represent this function with NDF as $f(x_1, \ldots, x_n, y) = 0$, where the field is defined for points $\mathbf{p} = (x_1, \ldots, x_n, y) \in \mathbb{R}^{n+1}$. If we want to find the value or multiple values of $y$ as for *fixed values* of $x_1, \ldots, x_n$, we can use the modified sphere tracing in Algorithm 2 by setting the initial point $\mathbf{p}_0 = (x_1, \ldots, x_n, 0)$ and the ray to $\mathbf{r} = (0, \ldots, 0, 1)^T$, see Fig. 9 (right). Note that since $f()$ can evaluate to 0 at multiple $y$ outputs for a given input, it can naturally represent multi-target data. Note that, while sphere tracing has been used exclusively for rendering, here we use it much more generally, to evaluate curves, manifolds or functions represented by NDF.

**Surface Normals and Differentiability of NDF:** For rendering images, normals at the surface are needed, which can be derived from UDF gradients. But since UDFs are not differentiable exactly at the surface, and NDF are trained to reproduce UDFs, gradients at the surface $f(\mathbf{p})$ will be inaccurate. However, UDFs are differentiable everywhere else, except at the cut locus $\mathcal{C}$. Far from the surface, the non-differentiability at the cut locus is not a problem in practice. Near the surface, it can be shown that the cut locus (points which are equidistant to at least two surface points) does not intersect a region of thickness $r_{\max}$ around the surface, if we can roll a closed ball $B_r$ of radius $r_{\max}$ inside and outside the surface, such that it touches all points in the surface [7] (see the supplementary for a visualization). When this condition is met, UDFs are differentiable ($C^1$) in a region $\mathcal{R}(\mathcal{S}) = \{\mathbf{x} \in \mathcal{R}^d \setminus \mathcal{S} \,|\, \mathrm{UDF}(\mathbf{x}, \mathcal{S}) < r_{\max}\}$, excluding points exactly on the surface. In practice, since NDF are learned, we compute gradients only at points in a region of $\epsilon = 5\mathrm{mm} < r_{\max}$ from the surface, which guarantees that we are sufficiently far without intersecting the cut locus – for surfaces of curvature $k < 1/\epsilon$. Hence, when rendering, we approximate the normal at the intersection point $\mathbf{q} \in \mathcal{S}$ by traveling back along the ray $\epsilon$ units and computing the gradient.

## 4  Experiments

We validate NDF on the task of *3D shape reconstruction* from sparse point-clouds, which is the main task we focus on in this paper. We first demonstrate that NDF can reconstruct **closed** surfaces on par with SOTA methods, and then we show that NDF can represent complex shapes with inner structures (**Complex Shape**). For comparisons we choose the Cars subset of ShapeNet [13] (and training and test split of [16]) for both experiments since this class has the greatest amount of inner-structures but can be also be closed (loosing inner-structure) which is required by prior implicit learning methods. In a second task, we show how NDF can be used to represent a wider class of surfaces, including **Functions and Manifolds**. Please refer to the supplementary for further experimental details and results. All results are reported on test data unseen during training.

**3D Shape Reconstruction of Closed Surfaces** In order to be able to compare to the state of the art [49, 15] + PSGN [24] +DMC [45], we train all on 3094 ShapeNet [13] cars pre-processed by [83] to be closed. This step looses all interior structures. We show reconstruction results when the input is 300 points, and 3000 points respectively. Point clouds are generated by sampling the closed Shapenet car models. In Fig. 5, we show that NDF can reconstruct closed surfaces with equal precision as SOTA [15]. In Table 1 we also compare our method quantitatively against all baselines and find NDF outperform all.

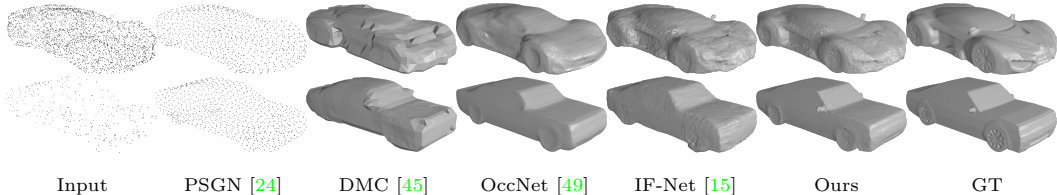

| Input | PSGN [24] | DMC [45] | OccNet [49] | IF-Net [15] | Ours | GT |

Figure 5: Comparison of methods trained on closed shapes (lost inner structure).

**3D Shape Reconstruction of Complex Shape.** To demonstrate the greater expressive power of our method, we train NDF to reconstruct complex shapes from sparse point cloud input. We train NDF respectively on shapes with inner-structures (*unprocessed* 5756 cars

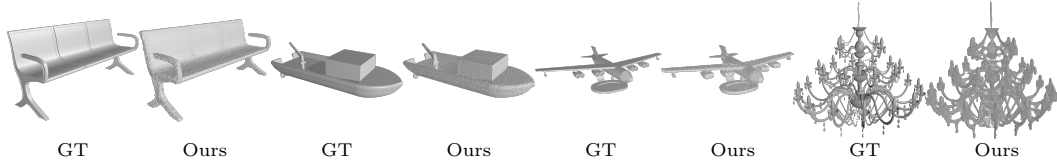

GT     Ours     GT     Ours     GT     Ours     GT     Ours

Figure 6: Reconstruction results on all classes of closed ShapeNet data from 3000 points trained with a single model. The quantitative Chamfer-$L_2 \downarrow$ results $\times 10^{-4}$ are: OccNet 4.0, PSGN 4.0, DMC 1.0, IF-Net 0.2 and NDF (ours) **0.05**.

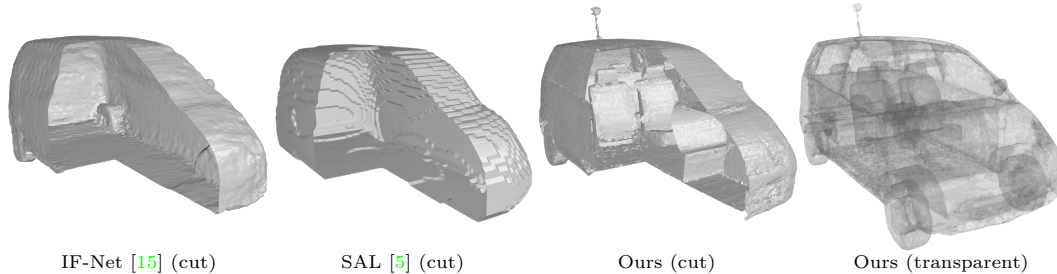

IF-Net [15] (cut)     SAL [5] (cut)     Ours (cut)     Ours (transparent)

Figure 7: Point cloud reconstruction of inner-structures on a test set car. Ours is the only method to successfully reconstruct full inner-structure. SAL and ours directly trained on raw data. IF-Net trained on closed data (without inner structure) for reference.

from ShapeNet [13], see Fig. 7) and open surfaces (307 garments from MGN [12], see Fig. 8) and 35 captured real world 3D spaces with open surfaces, see Fig. 1, all results are reported on test examples. This is only possible because NDF 1) are directly trainable on raw scans and 2) directly represent open surfaces in their output. This allows us to train on detailed ground truth data without lossy, artificial closing *and* to implicitly represent a much wider class of such shapes. We use SAL [5] and the *closed shape ground truth* as a baseline. As NDF, SAL does not require to closed training data. However, the final output is again an SDF prediction, which is limited to closed surfaces and can not represent the inner structures of cars, see Fig. 7. Moreover, in our experiments, when trained on multiple object classes jointly, SAL diverges from the signed distance solution and can not produce an output mesh. NDF can be trained to reconstruct all ShapeNet objects classes and again outperforms all baselines quantitatively (see Fig. 6). Table 1 shows that NDF are orders of magnitude more accurate than the closed ground truth and SAL, which illustrates that 1) a lot of information is lost considering just the outer surface, and 2) NDF can successfully recover inner structures and open surfaces.

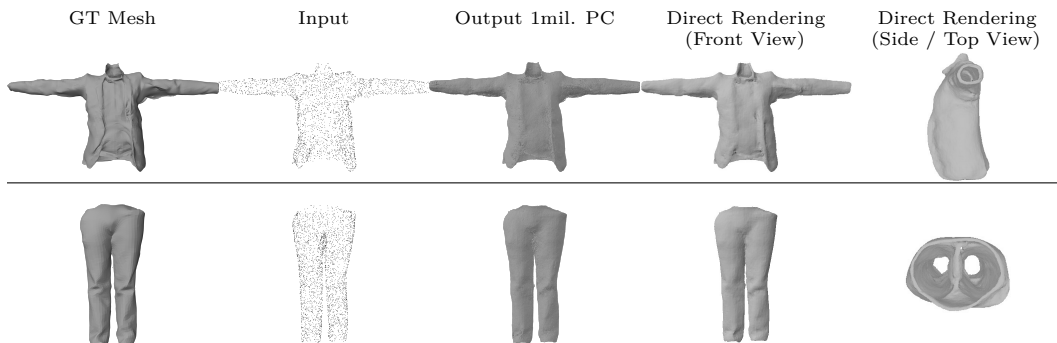

GT Mesh     Input     Output 1mil. PC     Direct Rendering (Front View)     Direct Rendering (Side / Top View)

Figure 8: Point Cloud Completion of garments, with 3000 input points, visualized using dense output point cloud generation (Alg 1) and direct renderings (Alg 2). Unlike all prior implicit formulations, our method can represent open surfaces. Data source is [12].

**Functions and Manifolds.** In this experiment, we show that unlike other IFL methods, NDF can represent mathematical functions and manifolds. For this experiment, we provide a sparse set of 2D points sampled from the manifold of a function as the input to NDF. We train a single Neural Distance Field on a dataset consisting of 1000 functions per type: linear function, parabola, sinusoids and spirals and $(x, y)$ point samples annotated with their

| | Chamfer-$L_2$ ↓ | | | | | | Chamfer-$L_2$ ↓ | | | |
|---|---|---|---|---|---|---|---|---|---|---|
| | 3000 Points | | 300 Points | | | | 10000 Points | | 3000 Points | |
| Input | 0.789 | 0.753 | 6.649 | 6.441 | Input | | 0.333 | 0.328 | 0.973 | 0.960 |
| PSGN [24] | 1.923 | 1.593 | 1.986 | 1.649 | - | | - | - | - | - |
| DMC [45] | 1.255 | 0.560 | 2.417 | 0.973 | - | | - | - | - | - |
| OccNet [49] | 0.938 | 0.482 | 1.009 | 0.590 | SAL [5] | | 6.39 | 5.79 | 7.39 | 5.44 |
| IF-Net [15] | 0.326 | 0.127 | 1.147 | 0.482 | W.GT | | 2.487 | 1.996 | 2.487 | 1.996 |
| Ours | **0.127** | **0.055** | **0.626** | **0.371** | Ours | | **0.074** | **0.041** | **0.275** | **0.086** |

Table 1: Results of point cloud completion for closed and unprocessed cars from 10000 points and 3000 and 300 points. Chamfer-$L_2$ results $\times 10^{-4}$. Left number shows the mean over Chamfer-$L_2$ scores, right the median. **Left Table:** Results training on pre-processed closed meshes. **Right Table:** Results training on raw scans. NDF can represent closed surfaces equally well than SOTA (left), and obtain a significant boost in accuracy when trained on raw data (right), because they can learn the inner structures.

ground truth unsigned distances in a bounding box from $-0.5$ to $0.5$ in both $x$ and $y$. We use a 80/20 random train and test split. Fig 9 demonstrates that NDF can interpolate sparse data-points and approximate the manifold or function. To obtain the surfaces from the predicted distance field, one could use the dense PC algorithm (Alg. 1) or the Ray-tracing algorithm (Alg. 2). In Fig. 9 we evaluate the NDF using the latter. To our knowledge this is the first application of sphere-tracing for regression.

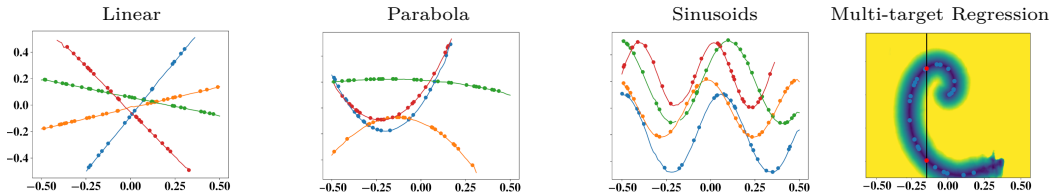

Figure 9: NDF can also represent and reconstruct arbitrary 2D surfaces at test time. This is interesting when the analytic form is unknown or non-existent. A single Neural Distance Field regresses linear, parabola and sinusoidal functions, as well as spiral curves. Dots are input samples and curves are regressions by our NDF network. NDF naturally extend to curves and we can effectively transfer ray tracing to 2D, for the purpose of regression. Right-most: We visualize the recovered distance field from the sparse spiral manifold and regress multiple curve values using root finding on a ray (Alg. 2).

**Limitations** Our method can produce dense point-clouds, but we rely on off-the-shelf methods for meshing them, which can be slow. Going directly from our NDF prediction to a mesh is desirable and we leave this for future work. Since our method is still not real time, a coarse-to-fine sampling strategy could significantly speed-up computation.

## 5 Discussion and Conclusion

We have shown how a simple change in the representation, from occupancy or signed distances to unsigned distances, significantly broadens the scope of current implicit function learning approaches. We introduced NDF, a new method which has two main advantages w.r.t. to prior work. First, NDF can learn directly from real world scan data without need to artificially close the surfaces before training. Second, and more importantly, NDF can represent a larger class of shapes including open surfaces, shapes with inner structures, as well as curves, manifold data and analytical mathematical functions. Our experiments with NDF show state-of-the-art performance on 3D point cloud reconstruction on ShapeNet. We introduce algorithms to visualize NDF either efficiently projecting points to the surface, or directly rendering the field with a custom variant of sphere tracing. We believe that NDF are an important step towards the goal of finding a learnable output representation that allows continuous, high resolution outputs of arbitrary shape.

**Acknowledgments** This work is funded by the Deutsche Forschungsgemeinschaft (DFG, German Research Foundation) - 409792180 (Emmy Noether Programme, project: Real Virtual Humans) and Google Faculty Research Award.

## Broader Impact

Real-world 3D data captured with sensors like cameras, scanners and lidar is often noisy and in the form of incomplete point clouds. NDFs are useful to reconstruct and complete such point clouds of complex objects and the 3D environment – this is relevant for AR or VR, autonomous vehicles, robotics, virtual humans, cultural heritage, quality assurance in fabrication.We show *state-of-the-art* results for this point cloud completion task. Application of NDFs for other 3D processing tasks, such as denoising and semantic segmentation can have further practical interest. NDFs are also of theoretical interest, in particular for representation learning of 3D shapes, and can open the door to new methods for manifold learning and multi-target regression. A potential danger of our method is that NDFs allow to learn from general world statistics to create complete 3D reconstructions from only sparsely or partially 3D captured environments, scenes and persons (e.g. gained from structure-from-motion techniques from images), which may violate personal or proprietary rights. The application of reconstruction algorithms, as ours, in safety relevant scenarios need to consider the risk of unreliable system predictions and should consider human intervention.

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
