[Supplementary Material]

# Supplementary Material: Neural Unsigned Distance Fields for Implicit Function Learning

Julian Chibane          Aymen Mir          Gerard Pons-Moll

## Abstract

In this supplementary document we provide details on training procedure and hyperparameters of our experiments in Sec. 1. In Sec. 2 we include results of point cloud completion of unprocessed scenes from a real-world 3D spaces dataset [4]. Finally, in Sec. 3 we provide further results of the experiments in our main paper.

## 1  Hyperparameters

**Network training**  For creating the training point samples with their ground truth UDF, we precompute 100.000 points in all experiments. For the ShapeNet experiments, we sample more points in the vicinity of the surface to capture details. For this we uniformly random sample a surface point $\mathbf{p}_{\mathcal{S}}$ on the ground-truth mesh and displace it with a Gaussian sample $\mathbf{d} \sim \mathcal{N}(0, \boldsymbol{\Sigma})$) to find a 3D point sample $\mathbf{p} := \mathbf{p}_{\mathcal{S}} + \mathbf{d}$. We use uncorrelated samples, that is, a diagonal covariance matrix $\boldsymbol{\Sigma} \in \mathbb{R}^{3 \times 3}$ with entries $\boldsymbol{\Sigma}_{i,i} = \sigma$. During training, sub-samples of 50.000 are used, with 1% of samples from $\sigma = 0.08$ to learn that far away points should have a clamped regressed distance $\delta = 10$cm, with 49% of samples from $\sigma = 0.02$ to learn to regress surface distances within $\delta$ and with 50% of samples from $\sigma = 0.003$ to learn to approximate the detailed surface boundary. We use the same values for the garments dataset. For scenes we use, 1% of samples from $\sigma = 0.16$, 49% of samples from $\sigma = 0.04$ and 50% of samples from $\sigma = 0.01$. For our function experiment we instead sub-sample 50.000 points uniformly in the bounding box from -0.5 to 0.5 in both x and y direction.
The cars and garments have been scaled such that their longest bounding box edge has length 1m and have been centered at (0,0,0), the scenes are kept in their metric system. To generate reconstructions of arbitrary size for scenes, we divide the 3D space into 2.5m$^3$ cubes and apply NDF in a convolutional sliding window approach.

We trained all baseline models according to their authors and used the convenient implementations from the authors of [3]. For IF-Nets and NDF the input point clouds are voxelized with resolution 128$^3$ for our comparison experiment on preprocessed watertight cars. For our experiment on the unprocessed full shapes we used the resolution of 256$^3$ for voxelization to better capture inner structures. For garments and scenes we also used 256$^3$. For optimizing the loss $\mathcal{L}_{\mathcal{B}}(\mathbf{w})$, we use the Adam optimizer with parameters $lr = 1e-4$, $betas = (0.9, 0.999)$, $eps = 1e-8$, $weight\_decay = 0$ in all experiments. To speed up training we initialized all networks with parameters gained by training on the full ShapeNet dataset. We trained the models until validation minimum was reached. All results in paper and supplementary material (quantitative and qualitative) are reported for test data (unseen during training). We used the common train and test split by [2] for all ShapeNet experiments.

**Hyperparameters of visualization algorithms**  On the ShapeNet dataset we use Algo. 1 to produce very dense point clouds from NDF predictions. We found that the point-to-surface mapping finds noise less surfaces with num_steps = 5 and doubled this number to num_steps = 10 in order to be robust against potential outliers. For rendering ShapeNet

results with Algo. 2 we used $\alpha = 0.6$, $\epsilon_1 = \epsilon_2 = 5$mm, that is, no taylor approximation and $\epsilon = 5$mm. For functions we use $\alpha = 1$, $\epsilon_1 = 0.03$, that is, a large first threshold in order to not overshoot, and use the Taylor approximation with $\beta = 0.1$ and $\epsilon_2 = 0.003$ to find the function value. As opposed to the regular shpere tracing, that always increses the distance from the initial point, using the Taylor approximation allows also to decrease it, in case of overshooting, that is, to approach the surface from both sides. For functions we do not evaluate the normals. For rendering garments we used $e_1 = e_2 = 2.6$mm and $\epsilon = 5$mm.

## 2 Scene Reconstruction

As an add-on, we show that NDFs can be trained for point cloud completion on unprocessed full scenes of the Gibson real-world 3D spaces dataset [4]. These scenes are created from RGB-D sensors and consist of open surfaces, (partially with holes) and thus, can not be used to train prior learned implicit function work. For training we use 35 scenes and apply NDFs in a sliding window scheme over the whole scene to obtain reconstructions of arbitrary large areas. The reconstructions for a full apartment is depicted in Fig. 2, for two floors of a house in Fig. 2 and for a living room in Fig. 2.

Figure 1: Unlike all prior implicit representations, NDFs able to represent complex open surfaces such as scenes. At the top we show the sparse input (150k points) point-cloud. In the middle we show our very dense reconstruction (3 Mio points) and at the bottom we show the ground-truth scan.

Figure 2: More results for scene completion. Top three rows show the first floor, bottom three the second floor of a house.

Figure 3: Scene reconstruction at room level. Our method preserves fine details such as the cushions of the sofa and the bowl on the table.

# 3 More Results

## 3.1 Garments

In Fig. 4, we show more results for point cloud completion on the garments dataset. For this experiment we use the garments dataset released by the authors of [1]. This dataset contains five garment-classes and we make use of all classes for our experiment. We use 297 garments for training, 20 for validation and 20 for testing. The input point clouds contain 3000 sample points. We visualize the output NDF using Algorithm 1 (dense PCs) and Algorithm 2 (direct renderings from multiple view points). Fig. 4 shows that NDFs can represent complex, open surfaces such as garments.

Figure 4: Point Cloud Completion of garments, with 3000 input points, visualized using dense output point cloud generation (Alg 1) and direct renderings (Alg 2). Unlike all prior implicit formulations, our method can represent open surfaces.

## 3.2 Renderings

In Fig. 5 and 6 we show more direct rendering results based on reconstructions of full, unprocessed ShapeNet cars from 10.000 input points. Each column represents one car and the rows show: input point cloud, depth image created with Algo. 2 by finding an approximate zero intersection, normal image (also created with Algo. 2), and subsequently, shaded images from different view points, created from normal images by computing angles to a light source in the scene.

Figure 5: Renderings of four cars, created with Algo. 2 from NDF reconstructions from 10.000 input points.

Figure 6: Renderings of four cars, created with Algo. 2 from NDF reconstructions from 10.000 input points.

## 3.3 Cars with inner structures

In Fig. 7, we show more results of the crucial difference between our NDF and prior implicit function learning: NDF can be learned on arbitrary surfaces without preprocessing and represent a broader class of shapes. Therefore, for cars, NDF are able to reconstruct inner structures, which are lost in the watertighting process of prior work. Since the surface areas of the cars with inner-structures are much larger than watertight shapes, we need to sample more points to reach comparable point densities. We show reconstructions from 3000 points for IF-Nets and 10000 points for NDFs.

| IF-Net (cut) | Ours (cut) | IF-Net (Transparent) | Ours (Transparent) |

Figure 7: Renderings of meshes reconstructed by NDF compared with renderings of meshes reconstructed by IF-Nets. Each row shows one car object and columns alternate between IF-Net and Ours. We show meshes where a part is cut to reveal the inner structure (left two columns) and meshes rendered with a transparent material (right two columns). While NDFs are able to reconstruct details inside the car, prior learned implicit functions are not.