[Reviews · NeurIPS 2020]

Review 1

Summary and Contributions: The paper proposes to regress unsigned distance function instead of signed distance as commonly did in implicit shape representation. It's shown that with the unsigned version of implicit functions, complicated 3D subjects with interior contexts could be faithfully represented.

Strengths: The idea of using unsigned disance field to represent open surfaces is an interesting idea. The paper also provides enough technical details on how to perform different kinds of rendering with the proposed unsigned implicit representation. It certainly holds technical value towards the ML & graphics community.

Weaknesses: I'm a bit confused by the claim that one of the benifit of NDF over SDF is the ability to also model functions and manifolds (Fig.1) Is it trivial that a function can be directly fitted by a network, what's does it mean to use unsigned distance field to represent a function? and what are the benifits? Similar question goes for manifold. The current experiment settings are limited to toy settings where near perfect 3D groundtruth are already given for training. What happens if the groundtruth is corrupted, incomplete, or sparse? Any comments or insights on how the proposed NDF could be applied to unsupervised learning with differentiable rendering?

Correctness: The intuition of using unsigned distance representation as well as the technical details in rendering appears to be correct to me. Some minor issues I had is discussed in above.

Clarity: The paper is mostly clearly written. It would be better if the author could explain a bit more why they view modeling functions / manifolds with NDF is prefered.

Relation to Prior Work: yes

Reproducibility: Yes

Additional Feedback:


Review 2

Summary and Contributions: Paper proposes an approach to produce un-signed distance field as a 3D shape representation for input sparse point cloud. This approach facilitate training on 3D dataset for which water tight meshes are hard to generate. Furthermore, this approach is also applicable for general curve, surface and manifold (spiral) approximation. The approach is simple and general, which promises wider applicability. Update: Authors have provided more experimental results on shapenet and comparison with SAL. Since baseline methods do no generate internal surfaces, chamfer distance will definitely be lower for NDF while using full meshes for ground truth. Therefore, the provided chamfer distance score for shapenet seems reasonable. I would be interested in an experiment where chamfer distance is only computed using external surfaces. Also, renderings are a bit noisy. I acknowledge the limitation of the paper that it doesn't fit in the classical rendering pipeline, I still think that the paper can be a good contribution to community, considering that it helps in generating internal surfaces and can work on non-water tight meshes (garments). As an after thought, how realistic is the setting of having just points without normals? Scanners give both points and normals albeit noisy. It seems bit unrealistic that you will have points sampled from internal surfaces, without any normals. If we have noisy points with normals, we can use the approaches from following papers: 1. Implicit geometric regularization for learning shapes 2. Implicit Neural Representations with Periodic Activation Functions Though both papers should be considered contemporary. I am willing to accept the paper if authors include detailed experiments on shapenet category, along with experiments on non water tight meshes like garments and open scenes.

Strengths: 1. Though, it is obvious how to produce un-signed distance field using implicit-function neural networks. It is less obvious that a simple gradient descent based method can help in recovering inherent surface as shown in algorithm-1,2. The approach is founded on the observation that you can recover the surface point by traversing in negative gradient direction of distance field, weighted by some constant. The observation is well founded. 2. The main contribution is how to extract the surface once you have a network that gives you reasonable un-signed distance field, which is explained in algorithm-1,2. 3. I appreciate the wider applicability of the approach, in terms of its usage in non-water-tight surfaces and approximation of manifolds, though more rigorous analysis is needed to establish the validity of latter. 4. Experiments are done on car category from shapenet and garments for shape reconstructions. Which definitely shows improvement over the presented baselines.

Weaknesses: 1. Unsigned distance field has been explored before in the context of surface approximation [A], which explores the scenario where signed distance is not available for training, which makes this approach less novel. A: SAL: Sign Agnostic Learning of Shapes from Raw Data 2. Since the experiments are only done on car and garments in the main paper and large scene reconstruction in the supplementary, it is hard to place this work in wider context of shape reconstruction tasks done in contemporary literature. For example, I would like to know how well it compares on other categories of shapenet. Is it better that signed distance based approaches? How does it compare with SAL [A] approach for shape reconstruction on water tight surfaces? I would prefer more details on comparison with SAl approach, may that be in supplementary material. If a researcher wants you use this method, can they expect better performance on water-tight as well as non water-tight surfaces?

Correctness: 1. The experiment section lacks comparison on all categories of shapenet, which makes it hard to judge whether this approach is applicable on wider categories. 2. Comparison with SAL is missing, which in my opinion is very relevant this work.

Clarity: 1. Paper is mostly well written. 2. There is a typo on line-76 (manifoldq -> manifold).

Relation to Prior Work: The paper covers most relevant works.

Reproducibility: Yes

Additional Feedback: 1. From what I understand, projection of point p to a surface point q as detailed in line 54 is only valid when norm of gradient is 1. If it is so, please either use different symbol or explicitly write it. 2. Please provide more information about run-time and num_steps. How does performance varies by changing num_steps. Can higher order derivatives be helpful in finding the surface points faster? 3. A good reconstruction describes the shape using small number of triangles. The fact that this algorithm can process millions of points and recover the inherent shape may not be desirable if you require millions of triangles to faithfully reconstruct the shape. This work needs a bit more analysis on how many initial points are needed to get a reasonable performance, along with tradeoffs. 4. Why normals computed away from the surface are good approximation of normals on the surface? How good is this approximation quantitatively? 5. Unlike signed distance function learning, does this approach guarantee f(p) to be zero on the surface up to certain tolerance? I can imagine the performance of this approach being sensitive to hyper-parameters. Does hyper parameter range vary much across different categories? In general, this paper can lead to good contribution to community if more detailed analysis and experiments are done. Paper does not inspire confidence in this particular approach.


Review 3

Summary and Contributions: The authors propose to represent 3D shapes using "NDFs" --- deep implicit unsigned distance functions. This representation admits a broader class of shapes than the recently popular SDF representations. Algorithms are provided to extract point clouds, meshes, or images from the learned implicit NDFs.

Strengths: Deep learning on 3D shapes in general and deep implicit representations for 3D geometry in particular are useful and exciting research topics. Expanding the class of shapes that can be represented in this fashion (e.g., not being limited to watertight manifolds) is a step towards more general 3D learning pipelines.

Weaknesses: The authors claim that their method has the benefits of (1) being able to learn from unoriented geometry, for which an SDF is not known/available and (2) being able to reconstruct non-manifold manifold geometry that does not have a well-defined SDF. With respect to (1), a more discussion and experimental comparison to "SAL: Sign Agnostic Learning of Shapes from Raw Data" is necessary. While the authors mention this work, they note that it still outputs SDFs. This should not preclude comparison experiments, e.g., on ShapeNet. On the other hand, with respect to (2), the authors don't sufficiently motivate the benefit of being able to learn implicit representations for non-manifold geometry---the examples under "Functions and Manifolds" are mainly toy examples. A lot of effort has been put into computing orientations for unoriented surfaces or point clouds, since algorithms for rendering, simulation, etc. often require consistent normals. By definition, the geometry represented by the proposed method is incompatible with these graphics pipelines. Additionally, there is insufficient details experimental details provided to evaluate the method compared to previous work. The authors only compare reconstruction quality on a single shape category (cars), and, moreover, very little information is provided about the set-up: what is the network architecture, learning rate, training time? Is there a test/train split, and are quantitive statistics provided on the test set?

Correctness: Overall, the claims made in the paper are valid, though a lot of technical details are missing that make it difficult to fully evaluate the experiments. The authors state that SDFs are "limited to 3D shape representations," whereas their method is more generic. This isn't true --- SDFs can certainly be used to capture 2D watertight manifolds (e.g., closed curves).

Clarity: The paper is generally clearly written. As mentioned above, more details about the specific set-up for the experiments is necessary for reproducibility and fair comparison.

Relation to Prior Work: The prior work is discussed sufficiently, and novel contributions are explicitly presented.

Reproducibility: No

Additional Feedback: Post-rebuttal update: Thank you to the authors for the clarifications. These have largely addressed my concerns with respect to comparisons with SAL. However, I am still not convinced that the contribution is sufficiently motivated. Generalizing SDFs to unsigned distances is not a particularly novel idea, and has even been tried in the context of deep learning. I'm not sure that demonstrated target application of modeling internal surfaces (e.g., cars) is a sufficiently convincing use case. I think the paper would be much stronger if it showed some of: novel theoretical results or ideas specific to unsigned learned implicit fields, application to modeling scenarios that truly require such a representation (e.g., garments), or extension of standard rendering pipelines to this representation.


Review 4

Summary and Contributions: This paper proposes an implicit representation for 3D geometries. Previous works apply signed distance field and are limited to water-tight surfaces. In contrast, this paper proposes unsigned distance field that can represent both water-tight and non-water-tight surfaces. The experiments show that the proposed representation can accurately represent complex geometries as well as curves and manifolds.

Strengths: 1. The introduction of signed distance field tackles a major limitation of previously commonly used signed distance field, and achieves more accurate reconstructions on non-water-tight surfaces such as vehicles with complex interior structures. 2. The paper proposes solutions to extracting dense point clouds from the learnt unsigned distance field, and present techniques for rendering surfaces and curves from the distance field. 3. The experiment results on representing complex interior structures of objects are impressive. Such a representation could be used in many other tasks in 3D reconstructions and neural rendering.

Weaknesses: 1. While the unsigned distance field is first applied in a deep learning setting, I would imagine that it has been well studied in traditional CG and CV. However, I didn't find many discussions on this in the paper. How do the algorithms used for point cloud extraction and ray tracing relate to previous works? Adding citations to previous works on related topics would help better position the paper. 2. Compared to SDF, the advantage of NDF is obvious. What are the potential disadvantages of NDF? Discussions on limitations would help readers to better understand the method. Overall I like the results presented in the paper, and the method also is interesting to me. The usage of NDF would be a good add-on to 3D deep learning.

Correctness: The paper is technically sound.

Clarity: The paper is well written.

Relation to Prior Work: More discussions to previous methods on unsigned distance field could be added.

Reproducibility: No

Additional Feedback: More details on the network architectures should be provided. Post-rebuttal: The rebuttal addresses my concerns. While signed distance field has its own problems, I believe it can be useful in some scenarios where waterlight surfaces are not applicable. So I keep my original score and agree with accepting the paper.

[Author Response · NeurIPS 2020]



| NDF (cars) | SAL[4] (cars) | GT | Ours | GT | Ours | GT | Ours | Steps vs Error |

Figure 1: Left - Comparison of NDFs with SAL. Middle three - NDF reconstructions for different ShapeNet categories. Right - Graph of: number of projection steps against Chamfer distance between reconst. and GT.

| Cars | | 10000 Points | | 3000 Points | |
| --- | --- | --- | --- | --- | --- |
| SAL[4] | | 6.39 | | 7.39 | |
| Ours | | **0.074** | | **0.275** | |

| Full ShapeNet | | OccNet | | PSGN | | DMC | | IF-Net | | Ours | |
| --- | --- | --- | --- | --- | --- | --- | --- | --- | --- | --- | --- |
| 3000 Points | | 4.0 | | 4.0 | | 1.0 | | 0.2 | | **0.05** | |

Table 1: Chamfer-$L_2$ ↓ results $\times 10^{-4}$. **Left Table:** Results of point cloud completion for unprocessed cars from 10.000 points and 3.000 points. **Right Table:** Results on full watertight ShapeNet from 3000 points.

We thank all reviewers for their useful feedback. Reviewers acknowledge that NDFs are "well founded",
"simple and general, which promises wider applicability"[**R2**], they "certainly hold technical value"[**R1**] and
all agree on their wide applicability. NDFs are interesting for researchers [**R4**] to build on it – we will make
all code and models available to the research community.

**Comparison with SAL[4] and full ShapeNet baselines** [**R2,R4**] We compare to SAL on the car class
of ShapeNet and we observe that it can reconstruct the watertight outer surface whereas NDF can reconstruct
also the inner structures. We tried training SAL on the full ShapeNet but it diverges from the signed distance,
so no meshes can be produced. We contacted the authors, they recommended us to train SAL on individual
classes. There is no evidence that it should work competitively on all classes together. In conclusion, SAL
is useful to watertight noisy meshes, and NDF is more flexible on the types of surfaces it can output. We
also added a comparison to other prior works on full ShapeNet. We conduct the experiment on watertight
surfaces as required by these works. We find that NDF outperforms all baselines.

**Unclear benefits of representing open surfaces** [**R1,R3**] Our NDFs 1.) are directly trainable on raw
scans and 2.) directly represent open surfaces in their output. This allows us to train on detailed ground truth
data without lossy watertighting *and* to implicitly represent a much wider class of shapes. All reviewers agree
on NDFs wider applicability. Note, our paper is not limited to toy examples [**R1,R3**], we show examples for
complex interior structures of cars, reconstruction of garments, completion of huge multi-object scenes.(Scenes
in Supplementary) Reviewers [**R1,R2,R4**] acknowledge our faithful 3D reconstruction results.
As an add-on, and to inspire future research, we also show toy examples for 2D representation of curves,
functions and manifolds, demonstrating NDFs are not limited to closed 3D surfaces. This is interesting [**R1**],
when a curve with multiple $y$ values for one $x$ should be represented, as in contrast to NDFs, a NN trained to
predict $y$ from $x$, would create an average.

**Experiment details**[**R3,R4**] *"Is there a test/train split, and are quantitative statistics provided on the test*
*set?" "what is the network architecture, learning rate?"*[**R3**] Please see the Supplementary for the answer. To
make the paper self-contained, we added more details from the reference architecture of [13] that we build on.
We follow the suggestion and added training time. We also transferred key information into the main paper.

**NDF Properties** [**R2**] *More information about run-time and num_steps* We added this ablation see figure
above, each num_ step takes 3.7 sec on a Tesla V100 for 1 Mio. points. *Why are normals computed in*
*the surrounding a good approximation?* A UDF is non-differentiable at 0 but its surrounding gives normal
information: if we move a small step away from the surface in normal direction, the closest point is unchanged
and we estimate the correct normal. If we move away in not normal direction, we rely on local smoothness of
the normal field.

**NDF Limitations** [**R4,R2**] 1.) Unlike SDFs, no standard contouring algorithm like Marching Cubes exists
for NDFs. Using BPA[8] for meshing on a predicted dense point cloud can lead to large amounts of triangles
and is slow to compute. 2.) As for all implicit models, a lot of points have to be predicted at inference.
To save time for NDF (and SDF) inference, our point to surface projection can be explored to do coarse
predictions that give distance and direction on where to do finer sampling.

**Works on unsigned distance fields in traditional CG/CV?** [**R4**] We investigated traditional literature
(lines 52-56 for point to surface mapping and in lines 197-203 for rendering). Distance fields are predominantly
used as *signed* distance fields. We added more related classical papers on shape matching and sculpting.

**Other suggestions and comments** We focused on common and most pressing concerns in the rebuttal.
We value all other suggestions by reviewers and incorporated them into the paper.

[Meta-Review · NeurIPS 2020]

The paper was extensively discussed post-rebuttal with points ranging from the concerns about comparisons with SAL. The reviews were split in the end. The AC examined the paper, the reviews, and the rebuttal. The AC is inclined to agree with the reviewers arguing to accept. In particular: there are subtle but important differences with SAL (which the authors should make extremely clear in the paper -- many readers, like the reviewers -- will ask about this); the paper serves as a good contribution in an area of intense interest for the field. The AC urges the authors to include all the information from the rebuttal into the camera ready, including (if possible) addressing the negative comments of R3.